# Epigenetic reprogramming by TET enzymes impacts co-transcriptional R-loops

João C Sabino[1], Madalena R de Almeida[1], Patrícia L Abreu[1], Ana M Ferreira[1], Paulo Caldas[2,3], Marco M Domingues[1], Nuno C Santos[1], Claus M Azzalin[1], Ana Rita Grosso[2,3], Sérgio Fernandes de Almeida[1]*

[1]Instituto de Medicina Molecular João Lobo Antunes, Faculdade de Medicina, Universidade de Lisboa, Lisbon, Portugal; [2]Associate laboratory i4HB – Institute for Health and Bioeconomy, NOVA School of Science and Technology, Universidade Nova de Lisboa, Caparica, Portugal; [3]UCIBIO-REQUIMTE, Applied Molecular Biosciences Unit, Department of Life Sciences, NOVA School of Science and Technology, Universidade Nova de Lisboa, Lisbon, Portugal

*For correspondence: sergioalmeida@fm.ul.pt

Competing interest: The authors declare that no competing interests exist.

**Abstract** DNA oxidation by ten-eleven translocation (TET) family enzymes is essential for epigenetic reprogramming. The conversion of 5-methylcytosine (5mC) into 5-hydroxymethylcytosine (5hmC) initiates developmental and cell-type-specific transcriptional programs through mechanisms that include changes in the chromatin structure. Here, we show that the presence of 5hmC in the transcribed gene promotes the annealing of the nascent RNA to the template DNA strand, leading to the formation of an R-loop. Depletion of TET enzymes reduced global R-loops in the absence of gene expression changes, whereas CRISPR-mediated tethering of TET to an active gene promoted the formation of R-loops. The genome-wide distribution of 5hmC and R-loops shows a positive correlation in mouse and human stem cells and overlap in half of all active genes. Moreover, R-loop resolution leads to differential expression of a subset of genes that are involved in crucial events during stem cell proliferation. Altogether, our data reveal that epigenetic reprogramming via TET activity promotes co-transcriptional R-loop formation, disclosing new mechanisms of gene expression regulation.

## Editor's evaluation

The study shows a correlation between 5hmC and R loops in mES cells and human HEK293 cells depleted of the TET enzymes Tet1, Tet2 and Tet3 that convert 5mC into 5hmC. The data presented are clearly of significant interest in providing new insight into the potential role of 5hmc DNA in specific transcriptional processes. This implies that 5hmC containing DNA has specific epigenetic features beyond a simple intermediate in interconversion between repressive 5mC and active C DNA.

## Introduction

During transcription, the nascent RNA molecule can hybridize with the template DNA and form a DNA:RNA hybrid and a displaced DNA strand. These triple-stranded structures, called R-loops, are physiologically relevant intermediates of several processes, such as immunoglobulin class-switch recombination and gene expression (*García-Muse and Aguilera, 2019*). However, nonscheduled or persistent R-loops constitute an important source of DNA damage, namely, DNA double-strand

breaks (DSBs) (*García-Muse and Aguilera, 2019*). To preserve genome integrity, cells possess diverse mechanisms to prevent the formation of R-loops or resolve them. R-loop formation is restricted by RNA-binding proteins and topoisomerase 1, whereas R-loops are removed by ribonucleases and helicases (reviewed in *García-Muse and Aguilera, 2019*). The ribonuclease H enzymes RNase H1 and RNase H2 degrade R-loops by digesting the RNA strand of the DNA:RNA hybrid. DNA and RNA helicases unwind the hybrid and restore the double-stranded DNA (dsDNA) structure. Several helicases unwind R-loops at different stages of the transcription cycle and in distinct physiological contexts (*García-Muse and Aguilera, 2019*). For instance, we previously reported that the DEAD-box helicase 23 (DDX23) resolves R-loops formed during transcription elongation to regulate gene expression programs and prevent transcription-dependent DNA damage (*Sridhara et al., 2017*). Intrinsic features of the transcribed DNA also influence its propensity to form R-loops. The presence of introns, for instance, prevents unscheduled R-loop formation at active genes (*Bonnet et al., 2017*). An asymmetrical distribution of guanines (G) and cytosines (C) nucleotides in the DNA duplex also influences R-loop propensity, with an excess of Cs in the template DNA strand (positive G:C skew) favoring R-loop formation (*Ginno et al., 2013*). Moreover, chromatin and DNA features such as histone modifications, DNA-supercoiling, and G-quadruplex structures also affect R-loop establishment (*García-Muse and Aguilera, 2019*). R-loops can also drive chromatin modifications. Promoter-proximal R-loops enhance the recruitment of the Tip60–p400 histone acetyltransferase complex and inhibit the binding of polycomb-repressive complex 2 and histone H3 lysine-27 methylation (*Chen et al., 2015*). R-loops formed over G-rich terminator elements promote histone H3 lysine-9 dimethylation, a repressive mark that reinforces RNA polymerase II pausing during transcription termination (*Skourti-Stathaki and Proudfoot, 2014*; *Chédin, 2016*; *Skourti-Stathaki et al., 2014*).

Besides affecting histone modifications, R-loops also act as barriers against DNA methylation spreading into active genes (*Ginno et al., 2013*; *Ginno et al., 2012*). DNA methylation, namely, 5-methylcytosine (5mC), results from the covalent addition of a methyl group to the carbon 5 of a C attached to a G through a phosphodiester bond (CpG) (*Karpf, 2013*). The activity of DNA methyltransferase (DNMT) enzymes makes 5mC widespread across the mammalian genome where it plays major roles in imprinting, retrotransposon silencing, and gene expression (*Greenberg and Bourc'his, 2019*). More than 70% of all human gene promoters contain stretches of CpG dinucleotides, termed CpG islands (CGIs), whose transcriptional activity is repressed by CpG methylation (*Greenberg and Bourc'his, 2019*; *Weber et al., 2007*). R-loops positioned near promoters of active genes maintain CGIs in an unmethylated state (*Ginno et al., 2012*), likely by reducing the affinity of DNMT1 binding to DNA (*Grunseich et al., 2018*), or recruiting ten-eleven translocation (TET) methylcytosine dioxygenases (*Arab et al., 2019*).

The TET enzyme family members share the ability to oxidize 5mC to 5-hydroxymethylcytosine (5hmC) (*Pastor et al., 2013*; *Tahiliani et al., 2009*). 5hmC is a relatively rare DNA modification found across the genome much less frequently than 5mC (*Mendonca et al., 2014*). Genome-wide, 5hmC is more abundant at regulatory regions near transcription start sites (TSSs), promoters, and exons, consistent with its role in gene expression regulation (*Wu et al., 2011*). The levels of 5hmC are enriched at active promoter regions, as observed upon activation of neuronal function-related genes in neural progenitors and neurons (*Pastor et al., 2013*; *Hahn et al., 2013*). 5hmC has the potential to modify the DNA helix structure by favoring DNA-end breathing motion, a dynamic feature of the protein–DNA complexes thought to control DNA accessibility (*Mendonca et al., 2014*). Moreover, 5hmC weakens the interaction between DNA and nucleosomal H2A-H2B dimers, facilitating RNA polymerase II elongation, and diminishes the thermodynamic stability of the DNA duplex (*Mendonca et al., 2014*). While 5mC increases the melting temperature, 5hmC reduces the amount of energy needed to separate the two strands of the DNA duplex (*Leavitt et al., 2015*; *Wanunu et al., 2011*). Molecular dynamics simulations revealed that the highest amplitude of GC DNA base-pair fluctuations is observed in the presence of 5hmC, whereas 5mC yielded GC base pairs (bp) with the lower amplitude values (*Wanunu et al., 2011*). The presence of 5hmC destabilizes GC pairing by alleviating steric constraints through an increase in molecular polarity (*Wanunu et al., 2011*).

Because features that destabilize the DNA duplex, such as supercoiling or G-quadruplexes, are known to facilitate nascent RNA annealing with the template DNA strand, we reasoned that 5hmC may favor R-loop formation. Here, we show that 5hmC promotes R-loop formation during in vitro transcription of DNA templates. In vivo, depletion of TET enzymes reduces R-loop levels, whereas

targeting the enzyme to an active gene drives R-loop formation. Analysis of genome-wide distribution profiles shows a positive correlation between 5hmC and R-loops in mouse embryonic stem (mES) and in human embryonic kidney 293 (HEK293) cells, with a clear overlap of 5hmC and R-loops in approximately half of all active genes. We also show that 5hmC-rich regions are characterized by increased levels of phosphorylated histone H2AX (γH2AX), a marker of DNA damage. Finally, by determining the pathways more significantly affected by R-loops formed at 5hmC loci, we disclose novel links between R-loops and gene expression programs of stem cells.

## Results

### Transcription through 5hmC-rich DNA favors R-loop formation

To assess the impact of cytosine methylation on R-loop formation, we performed in vitro T7 transcription of DNA fragments containing either native or modified cytosine deoxyribonucleotides (dCTPs). We synthesized three distinct DNA transcription templates, each composed of a T7 promoter followed by a 400 bp sequence containing a genomic region prone to form R-loops in vivo (*Sridhara et al., 2017*; *Skourti-Stathaki et al., 2014*). Two of these sequences (*ACTB* P1 and *ACTB* P2) are from the transcription termination region of the human β-actin coding gene (*ACTB*); the third sequence is from the human *APOE* gene. The DNA templates for the in vitro transcription reactions were generated by PCR-amplification in the presence of dNTPs containing either native C, 5mC, or 5hmC (*Figure 1A*). Successful incorporation of dCTP variants was confirmed by immunoblotting using specific antibodies against each variant (*Figure 1B*). The formation of R-loops during the in vitro transcription reactions was inspected by blotting immobilized RNAs with the S9.6 antibody (S9.6 Ab), which binds DNA:RNA hybrids (*Figure 1C*). To increase the specificity of hybrid detection, all samples were treated with RNase A in high-salt conditions in order to digest all RNA molecules except those engaged in R-loops. The specific detection of DNA:RNA hybrids was confirmed by blotting transcription reaction products previously digested with RNase H (*Figure 1C*). In agreement with our hypothesis that 5hmC favors R-loops, increased amounts of DNA:RNA hybrids were detected in samples derived from in vitro transcription of 5hmC-rich *ACTB* P1, *ACTB* P2, and *APOE* DNA templates (*Figure 1D*). To exclude the possibility that our results were biased by an inherent preference of the S9.6 Ab for hybrids containing 5hmC, we performed electrophoretic mobility shift assays (EMSAs) using the S9.6 Ab and DNA:RNA hybrid substrates of the same sequence but containing C, 5mC, or 5hmC. The S9.6 Ab was able to delay the run of the three substrates with similar kinetics, indicating that the Ab equally recognizes DNA:RNA hybrids formed with any of the three C variants (*Figure 1—figure supplement 1*).

We then performed atomic force microscopy (AFM) to directly visualize R-loop structures obtained in the in vitro transcription reactions (*Figure 1E*). R-loops were identified as previously described (*Carrasco-Salas et al., 2019*; *Klinov et al., 1998*). Each individual DNA molecule establishing an R-loop structure in the AFM images was assigned manually. The frequency of these structures formed in the presence of C, 5hmC, or 5mC DNA templates was measured and normalized against the frequency formed in RNase H-treated samples (*Figure 1E*). In agreement with the hypothesis that transcription of 5hmC-rich DNA templates results in increased R-loop formation, AFM data revealed that R-loop structures are more frequently formed in the presence of 5hmC.

To investigate if the DNA modification impacts in vitro transcription levels, we measured RNA synthesis from DNA templates containing unmodified C, 5hmC, or 5mC (*Figure 1—figure supplement 2*). These data show that the T7 polymerase is highly sensitive to DNA modifications since replacing C by either 5hmC or 5mC significantly decreased the transcript levels in vitro. On one side, detecting more R-loops on a lower-transcription levels setting (i.e., 5hmC-rich templates) further strengthens our hypothesis that 5hmC increases R-loop formation. However, we cannot draw any conclusions regarding the impact of 5mC on R-loop formation as a putative effect on R-loop levels could be masked by the significantly altered transcription. To clarify this aspect and further test our model, we continued our study with experiments performed in vivo.

### TET enzymatic activity impacts endogenous R-loop levels

To test whether the 5hmC DNA modification induces R-loop formation in vivo, we quantified R-loop levels in mES cells after depletion of *Tet1*, *Tet2,* and *Tet3*. Despite the significant reduction in Tet enzymes (*Figure 2—figure supplement 1A*), the levels of 5hmC were not significantly affected by

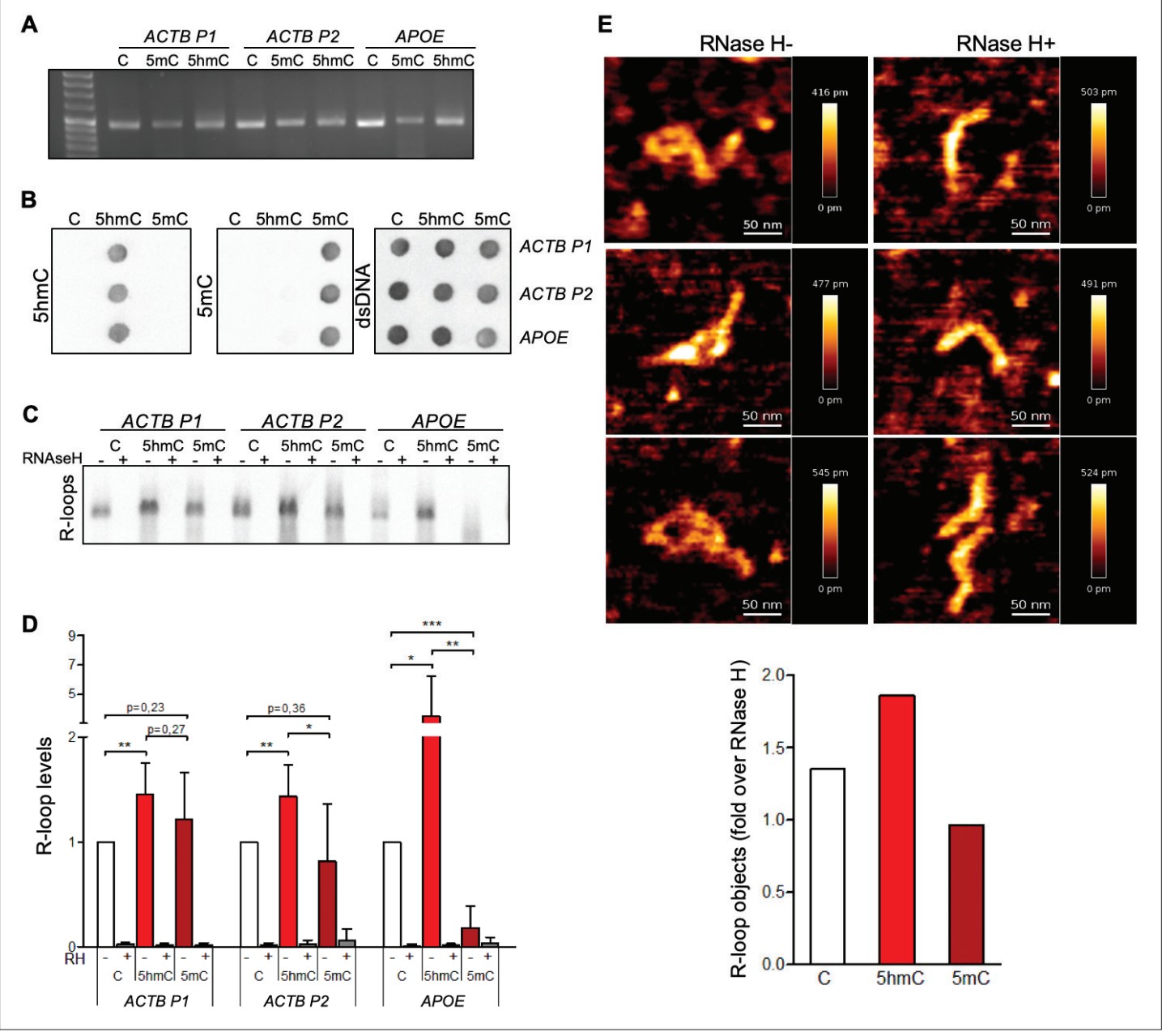

**Figure 1.** 5-Hydroxymethylcytosine (5hmC) favors co-transcriptional R-loop formation. (**A**) Native or modified deoxyribonucleotides (dCTPs) were incorporated upon PCR amplification into DNA fragments with sequences from the transcription termination region of *ACTB* (*ACTB* P1 and *ACTB* P2) or *APOE*. (**B**) Incorporation of dCTP variants confirmed by immunoblotting using specific antibodies against 5-methylcytosine (5mC), 5hmC, and double-stranded DNA (dsDNA). (**C**) R-loops formed upon in vitro transcription reactions were detected by immunoblotting using the S9.6 antibody. RNase H-treated in vitro transcription reaction products (RH+) serve as negative controls. All data are representative of seven independent experiments with similar results. (**D**) S9.6 immunoblots were quantified and the R-loop levels normalized against the levels detected in the reaction products of DNA templates containing native C. Data represent the mean and standard deviation (SD) from seven independent experiments. *p<0.05, **p<0.01, and ***p<0.001, two-tailed Student's *t*-test. (**E**) In vitro transcription reaction products of *ACTB* P2 templates were visualized using atomic force microscopy (AFM). R-loop structures obtained from 5hmC-containing *ACTB* P2 transcription in the absence (RH-) or presence (RH+) of RNase H are shown. R-loops present in the transcription reaction products of C, 5mC, or 5hmC-containing *ACTB* P2 templates were counted in a minimum of 80 filaments observed in three individual AFM experiments.

The online version of this article includes the following figure supplement(s) for figure 1:

**Figure supplement 1.** Cytosine modifications do not affect the detection of DNA:RNA hybrids by the S9.6 Ab.

**Figure supplement 2.** In vitro transcription levels of different DNA templates.

*Tet1* or *Tet2* depletion (*Figure 2A*, *Figure 2—figure supplement 1B*). In contrast, depletion of *Tet3* resulted in a significant loss of 5hmC, an effect that was exacerbated by the simultaneous depletion of the three enzymes (*Figure 2A*, *Figure 2—figure supplement 1B*). No significant changes were observed in 5mC levels (*Figure 2A*, *Figure 2—figure supplement 1B*). This finding suggests that there is a partial redundancy in the activity of the three Tet enzymes in mES cells. The loss of Tet1 or Tet2 – but not of Tet3 – is compensated by the remaining Tets. In agreement with the hypothesis that 5hmC promotes R-loop formation, dot-blot hybridization of total cellular nucleic acids using the S9.6 Ab revealed reduced endogenous R-loop levels in mES cells after depletion of Tet3 and after co-depletion of the three Tet enzymes (*Figure 2B*, *Figure 2—figure supplement 1B*). We also measured R-loop levels upon RNAi depletion of the Tet enzymes in NIH-3T3 mouse fibroblasts (*Figure 2—figure supplement 2A*). As observed in mES cells, a significant reduction of 5hmC, but not 5mC, was obtained upon depletion of Tet3 and of the three Tets in mouse fibroblasts (*Figure 2—figure supplement 2B, C*). The triple knockdown of the Tet enzymes significantly reduced the global levels of R-loops in mouse fibroblasts, whereas Tet3 depletion in these cells had a minor impact in R-loops (*Figure 2—figure supplement 2B and D*). This effect was further confirmed by measuring R-loops formed at selected active genes by DNA:RNA immunoprecipitation (DRIP) in mES cells (*Figure 2C*) and mouse fibroblasts (*Figure 2—figure supplement 2E*). The DRIP assays confirmed that R-loops are less abundant upon depletion of Tet enzymes. Importantly, simultaneous depletion of the three enzymes did not affect the expression levels of the analyzed genes in mES cells and mouse fibroblasts (*Figure 2D*, *Figure 2—figure supplement 2F*). These data suggest that the activity of Tet enzymes promotes the formation of R-loops in the absence of changes in transcription levels.

Next, we employed a modified CRISPR-based system to target TET enzymatic activity to specific loci (*Liu et al., 2016*). We used a pool of three specific guide RNAs (gRNAs) to direct a catalytically inactive Cas9 nuclease fused to the catalytic domain of TET1 (dCas9-TET1) to the last exon of the *APOE* gene in human osteosarcoma (U-2 OS) cells. As a control, dCas9 was fused to an inactive mutant version of the TET1 catalytic domain (dCas9-dTET1). Local enrichment of 5hmC following dCas9-TET1 targeting at the *APOE* locus was confirmed by DNA immunoprecipitation using antibodies specific for 5mC or 5hmC-modified nucleotides (*Figure 2E*). The highest levels of 5hmC were detected at the gene segment adjacent to the gRNAs-target region. R-loop levels detected by DRIP peaked significantly at the gRNAs-target and in the downstream region, upon tethering of dCas9-TET1 but not of dCas9-dTET1 (*Figure 2F*). These differences were not caused by changes in *APOE* gene expression levels (*Figure 2G*). The increased levels of R-loops detected far from the dCas9-TET1 target site are consistent with the view that R-loops have the capacity to extend from their inception locus. Accordingly, R-loops can be up to several hundred base pairs long and may extend over the entire gene body of shorter and/or highly transcribed genes (*Sanz et al., 2016*; *Chen et al., 2017*). Collectively, these data suggest that editing 5hmC density by changing the expression levels or the genomic distribution of TET enzymes influences R-loop formation in cells.

## 5hmC and R-loops overlap genome-wide at transcriptionally active genes

To further inspect the link between 5hmC and R-loops, we performed computational analyses of 5hmC antibody-based DNA immunoprecipitation (hMeDIP-seq) and DNA:RNA immunoprecipitation (DRIP-seq) datasets from mES and HEK293 cells (*Chen et al., 2015*; *Matarese et al., 2011*; *Jin et al., 2014*; *Nadel et al., 2015*). To assess individual genome-wide distribution profiles, R-loops density was probed over fixed windows of ±10 kbp around the 5hmC peaks (*Figure 3A*, *Figure 3—figure supplement 1A*). The resulting metagene plots and heatmaps revealed a marked overlap between 5hmC-rich loci and R-loops. This overlap is also evident in the individual distribution profiles of 5hmC and R-loops along two long regions of chromosome 17 (*Figure 3—figure supplement 2*). Despite the distinct distribution patterns of 5hmC (well-defined peaks) and R-loops (reads spanning genomic regions with highly heterogeneous lengths, ranging between a few dozen to over 1 kb *Chen et al., 2015*), we could obtain a statistically significant Pearson correlation coefficient between both ($p<0.05$) (*Figure 3B*, *Figure 3—figure supplement 1B*). Furthermore, approximately half of all R-loops detected genome-wide occurred at 5hmC-containing loci (*Figure 3C*, *Figure 3—figure supplement 1C*). Notably, we observed an overlap between 5hmC and R-loops in 6839 (51%) out of the 13,288 actively expressed genes (*Figure 3D*), a feature illustrated in the individual profiles of mouse and

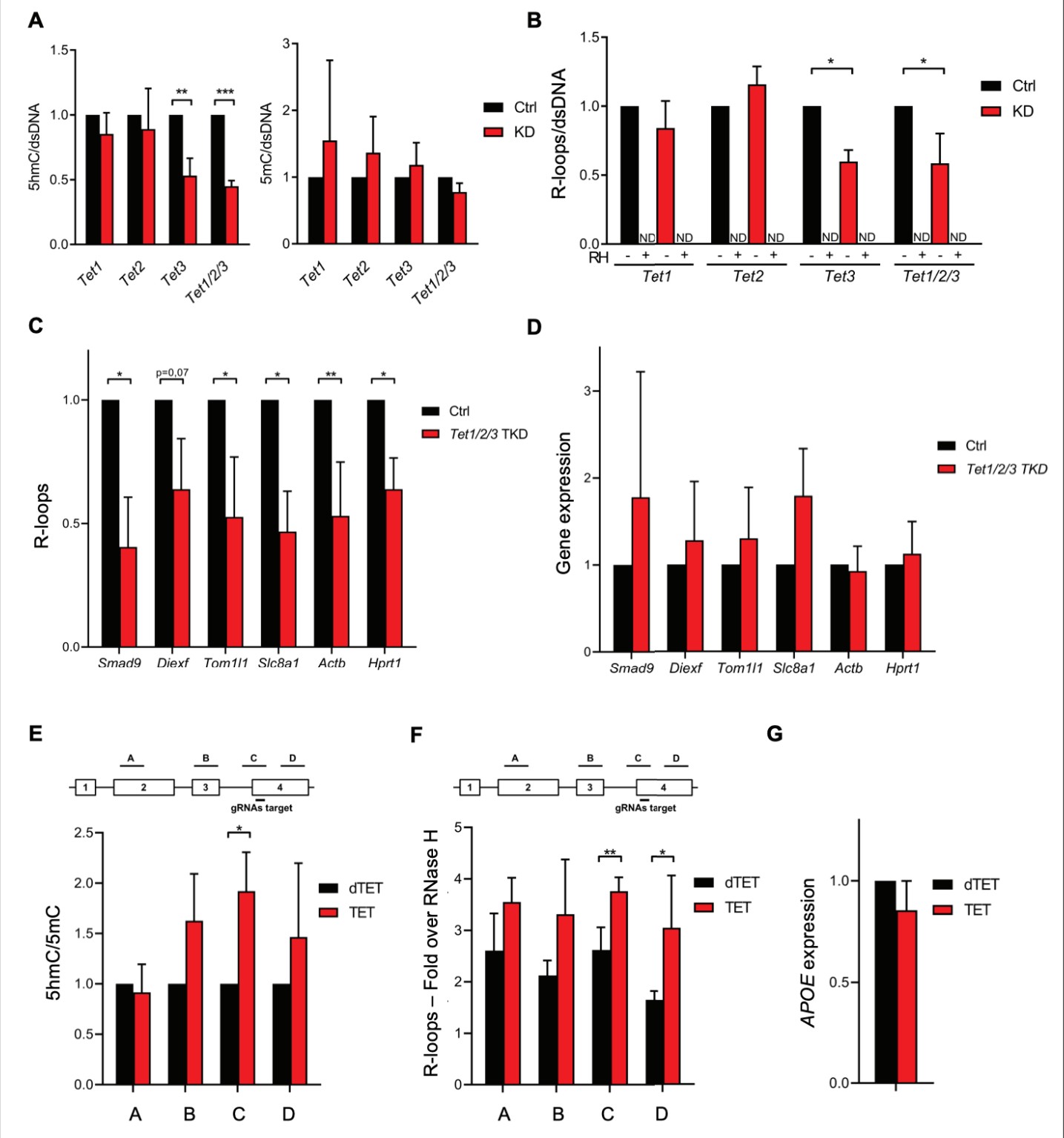

**Figure 2.** Ten-eleven translocation (TET) enzymatic activity impacts R-loop levels. Quantification of 5-hydroxymethylcytosine (5hmC) and 5-methylcytosine (5mC) (**A**) and R-loops (**B**) dot blots of *Tet1*, *Tet2*, *Tet3* single KD, and of *Tet1/2/3* triple KD mouse embryonic stem (mES) cells. Dot blots are shown in *Figure 2—figure supplement 1B*. Data were normalized against dsDNA levels. *p<0.05, **p<0.01, ***p<0.001, two-tailed Student's *t*-test. ND, not detected. (**C**) R-loop levels assessed by DNA:RNA immunoprecipitation (DRIP) in *Tet1/2/3* triple KD mES cells. Data were normalized against RNase H-treated samples. *p<0.05, **p<0.01, two-tailed Student's *t*-test. (**D**) Transcription levels of the genes presented in (**C**) assessed by RT-qPCR. 5hmC/5mC (**E**) and R-loop (**F**) levels determined by (h)MeDIP or DRIP at four regions of the *APOE* gene upon tethering of dCas9-TET1 or dCas9-

*Figure 2 continued on next page*

*Figure 2 continued*

dTET1 to the last exon of *APOE* in U-2 OS cells. R-loop data were normalized against RNase H-treated samples. *p<0.05, **p<0.01, two-tailed Student's *t*-test. (**G**) *APOE* transcription levels upon targeting dCas9-TET1 or dCas9-dTET1 to the last exon of the gene in U-2 OS cells. Data shown are the mean and SD from at least three independent experiments.

The online version of this article includes the following figure supplement(s) for figure 2:

**Figure supplement 1.** Impact of *Tet* depletion in mouse embryonic stem (mES) cells in R-loops, 5-hydroxymethylcytosine (5hmC), and 5-methylcytosine (5mC).

**Figure supplement 2.** *Tet* depletion impacts R-loop formation in mouse fibroblasts.

human genes (*Figure 3E*, *Figure 3—figure supplement 1D*). Metagene profiles revealed very similar patterns of intragenic distribution, with both 5hmC and R-loops increasing towards the transcription termination site (TTS), where they reached maximum levels (*Figure 3F*). At the TSS, however, the 5hmC DNA modification was mostly absent, whereas R-loops were abundant. The detection of R-loop peaks at TSS regions is in agreement with previous studies (*Ginno et al., 2013*; *Ginno et al., 2012*) and implies that 5hmC is not necessary for co-transcriptional DNA:RNA hybridization and R-loop formation.

The observed overlap between 5hmC and R-loop peaks at the TTS raises the hypothesis that Tet activity may be involved in transcription termination by directing the formation of R-loops. Defects in transcription termination result in the accumulation of readthrough transcripts extending beyond the TTS (*Nojima and Proudfoot, 2022*). In agreement with a role in transcription termination, TET1-KO human ES cells displayed significantly higher levels of readthrough transcripts genome wide when compared to wt human ES cells (*Figure 3G*).

We then sought to simultaneously detect 5hmC and R-loops at the same loci in individual mES cells. We performed proximity ligation assays (PLAs) using S9.6 and anti-5hmC antibodies (*Figure 4A*). Control reactions without primary antibodies and with each antibody alone did not produce a significant signal. Staining of mES cells with S9.6 and anti-5hmC antibodies gave rise to a robust PLA signal scattered throughout the nucleus, which was mostly lost after digestion of cells with RNase H (*Figure 4B*).

## 5hmC-rich loci are prone to DNA damage

Disruption of R-loop homeostasis is a well-described source of genomic instability (*García-Muse and Aguilera, 2019*). For instance, co-transcriptional R-loops increase conflicts between transcription and replication machineries by creating an additional barrier to fork progression (*Hamperl et al., 2017*; *Helmrich et al., 2013*). Such conflicts may cause DNA damage, including DSBs, which can be revealed using antibodies against γH2AX. Indeed, R-loops overlap with γH2AX-decorated chromatin at different locations such as TTS (*Hatchi et al., 2015*). We then sought to investigate if 5hmC creates conditions for DNA damage by promoting R-loop formation. We analyzed the genomic distribution of γH2AX by interrogating chromatin immunoprecipitation followed by sequencing (ChIP-seq) data from HEK293 cells (*Bunch et al., 2015*). The individual distribution profiles of γH2AX were analyzed over fixed windows of ±10 kbp around the 5hmC peaks detected in the same cells (*Figure 5A*). The resulting metagene plots revealed marked enrichment of γH2AX at 5hmC-rich loci. The genic distribution of 5hmC and R-loops along three different genes further showed co-localization of the two marks with γH2AX (*Figure 5B*). Analysis of γH2AX and 5hmC distribution within active genes revealed a low yet statistically significant Pearson correlation coefficient (p<0.05) (*Figure 5C*).

## R-loops formed at 5hmC-rich regions impact gene expression in mES cells

To gather insights into the functional impact of R-loops at 5hmC-rich DNA regions, we analyzed whole-transcriptome (RNA-seq) of mES cells overexpressing RNase H, a condition resulting in genome-wide loss of R-loops (*Chen et al., 2015*). Amongst the genes that were differentially expressed, we found that 64 and 48% of all downregulated and upregulated genes, respectively, displayed R-loops overlapping with 5hmC (*Figure 6A*). Pathway analysis revealed that these differentially expressed genes (*Supplementary file 1*) are involved in the mechanistic target of rapamycin (mTOR) (downregulated) and MYC (upregulated) signaling pathways (*Figure 6B and C*). mTOR and MYC are known to play

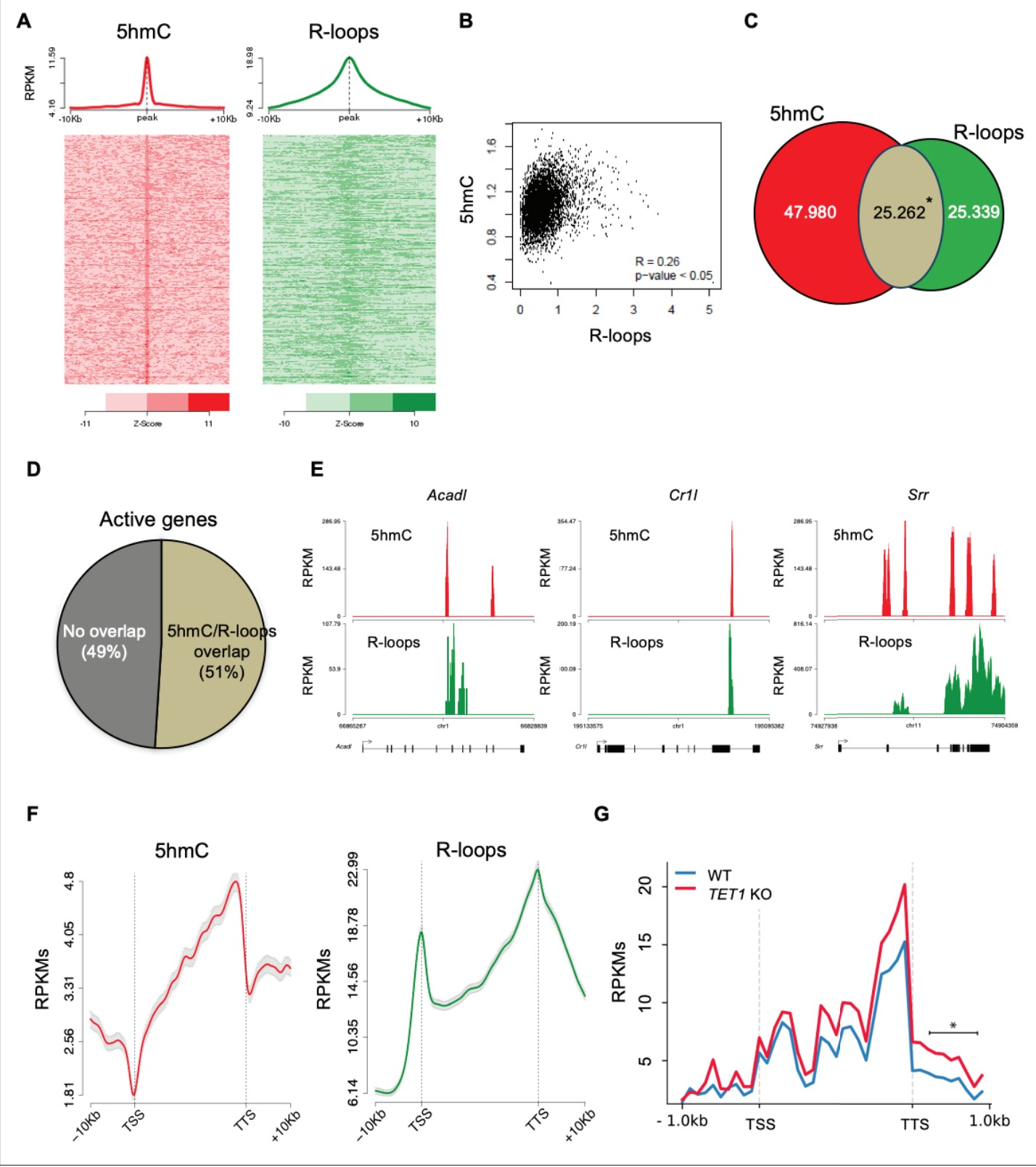

**Figure 3.** 5-Hydroxymethylcytosine (5hmC) and R-loops overlap in active genes of mouse embryonic stem (mES) cells. (**A**) Metagene and heatmap profiles of 5hmC and R-loops probed over fixed windows of ±10 kbp around the 5hmC peaks in expressed genes. (**B**) Pearson correlation coefficient between 5hmC and R-loops distribution within active genes (p<0.05). (**C**) Number of loci displaying 5hmC, R-loops, and overlapping 5hmC and R-loops. *Permutation analysis, p<0.05. (**D**) Percentage of active genes displaying overlapping 5hmC and R-loops. (**E**) Individual profiles of 5hmC and R-loop

*Figure 3 continued on next page*

*Figure 3 continued*

distribution along the *Acadl, Cr1l,* and *Srr* genes. Density signals are represented as reads per kilobase (RPKMs). (**F**) Metagene profiles of 5hmC and R-loops distribution in active genes. The gene body region was scaled to 60 equally sized bins, and ±10 kbp gene-flanking regions were averaged in 200 bp windows. TSS: transcription start site; TTS: transcription termination site. Density signals are represented as RPKMs, and error bars (gray) represent standard error of the mean. (**G**) Metagene profiles of genes showing transcription readthrough in wild-type and *TET1* KO human ES cells. All gene regions were scaled to 2000 bp (gene body) and divided in equal bins of 100 bp. 1000 bp regions averaged in 100 bp bins were added upstream the TSS and downstream the TTS region. *p<0.05, Mann–Whitney rank test.

The online version of this article includes the following figure supplement(s) for figure 3:

**Figure supplement 1.** Genome-wide analysis of 5-hydroxymethylcytosine (5hmC) and R-loops in HEK293 cells.

**Figure supplement 2.** Chromosomal distribution of R-loops and 5-hydroxymethylcytosine (5hmC).

opposite roles in establishing diapause, the temporary suspension of embryonic development driven by adverse environmental conditions (*Fenelon et al., 2014*), a stage that ES cells mimic when cultured in vitro. mTOR, a major nutrient sensor, acts as a rheostat during ES cell differentiation and reductions in mTOR activity trigger diapause (*Bulut-Karslioglu et al., 2016*). While overexpression of RNase H in mES cells did not reveal any significant changes in the cell cycle progression (*Figure 6—figure supplement 1A and B*), we observed a significantly decreased expression of genes related to pluripotency (*Pou5f1*) and germ layer commitment (*Sox17, Sox6, Dll1*) pathways (*Figure 6D*). These data support the view that R-loops formed upon TET epigenetic reprogramming regulate gene expression in stem cells.

## Discussion

In this study, we probed the hypothesis that 5hmC facilitates the co-transcriptional formation of noncanonical DNA secondary structures, known as R-loops. Data from in vitro transcription reactions and AFM provide direct evidence showing that transcription through 5hmC-rich DNA favors R-loop formation. By depleting TET enzymes in mES cells and fibroblasts, we demonstrate that TET activity increases cellular R-loop levels. Notably, the diminished levels of R-loops observed in TET-depleted cells did not result from impaired transcription, suggesting that 5hmC directly promotes R-loop formation. In agreement, tethering TET enzymes to a specific genomic locus using a CRISPR/Cas9-based system increases the levels of R-loops at the target locus.

As 5hmC is mostly absent from the TSS, other chromatin and DNA features (e.g., histone modifications, DNA-supercoiling or G-quadruplex structures; *García-Muse and Aguilera, 2019*) known to induce R-loop formation are likely to operate in these regions. In contrast, the robust overlap between R-loops and 5hmC at the TTS of active genes suggests a putative causal link. Mechanistically, 5hmC may impact R-loop formation by either destabilizing the DNA duplex or altering RNA polymerase II elongation rate. Indeed, 5hmC modifies the DNA helix structure by favoring DNA-end breathing motion, diminishes the thermodynamic stability of the DNA duplex, and destabilizes GC pairing (*Mendonca et al., 2014*; *Wanunu et al., 2011*). It also weakens the interaction between DNA and nucleosomal histones (*Mendonca et al., 2014*), which is thought to accelerate RNA polymerase II elongation but can also facilitate nascent RNA annealing with the template DNA strand favoring R-loop formation. Future studies will clarify which one of these mechanisms, if not all, contribute to the observed impact of 5hmC on R-loops.

Acting as a promoter of R-loops, well-established drivers of DNA damage (*García-Muse and Aguilera, 2019*), 5hmC may indirectly harm genome integrity. Indeed, we found that 5hmC-rich loci are hotspots for DNA damage genome-wide. While such unscheduled R-loops formed at 5hmC-rich loci may threaten genomic integrity, regulated formation of R-loops at specific 5hmC-decorated loci may exert important regulatory roles. Indeed, R-loops play diverse physiological functions (*García-Muse and Aguilera, 2019*), such as the regulation of gene expression. Our findings that genome-wide 5hmC and R-loops overlap robustly at the TTS of active genes and that TET-deficiency drives transcription readthrough support a model whereby TET enzymes act upstream of R-loop formation during transcription termination (*Skourti-Stathaki et al., 2011*).

TETs may play dual roles as both oncogenic and tumour suppressor genes, with the former arising as the consequence of altered expression levels or function, as observed in several cancers, such as triple-negative breast cancer (*Bray et al., 2021*; *Good et al., 2018*). In addition to altering the

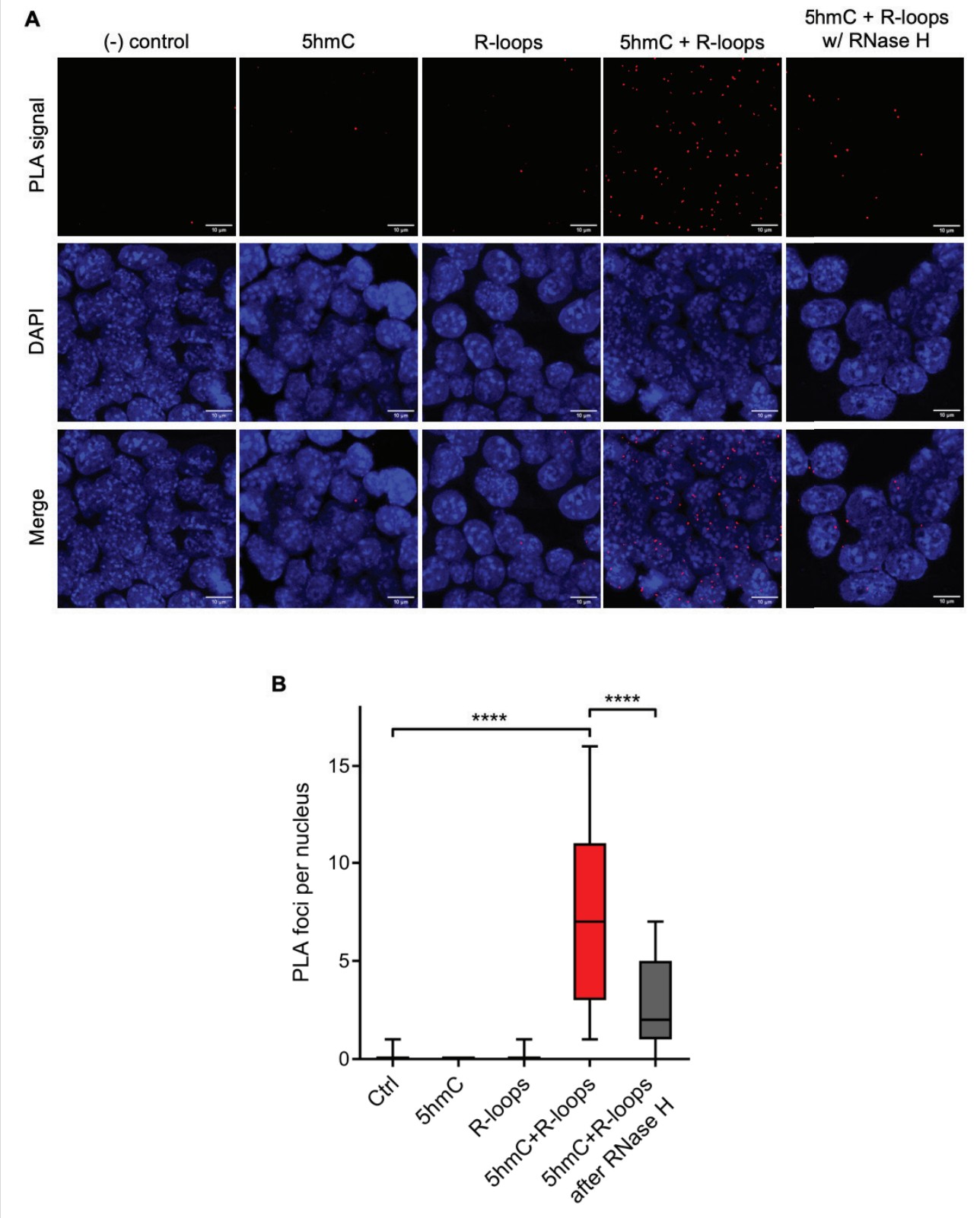

**Figure 4.** Simultaneous detection of 5-hydroxymethylcytosine (5hmC) and R-loops at the same genomic loci in individual mouse embryonic stem (mES) cells. (**A**) 5hmC and R-loops proximity ligation assay (PLA) foci in mES cells. DAPI was added to the mounting medium to stain DNA. Scale bars: 10 μm. Data are representative of at least three independent experiments with similar results. (**B**) Boxplot showing 5hmC/R-loops PLA foci per nucleus. Horizontal solid lines represent the median values, and whiskers correspond to the 10 and 90 percentiles. A minimum of 300 cells from at least three independent experiments was scored for each experimental condition. ****p<0.0001, Mann–Whitney rank test.

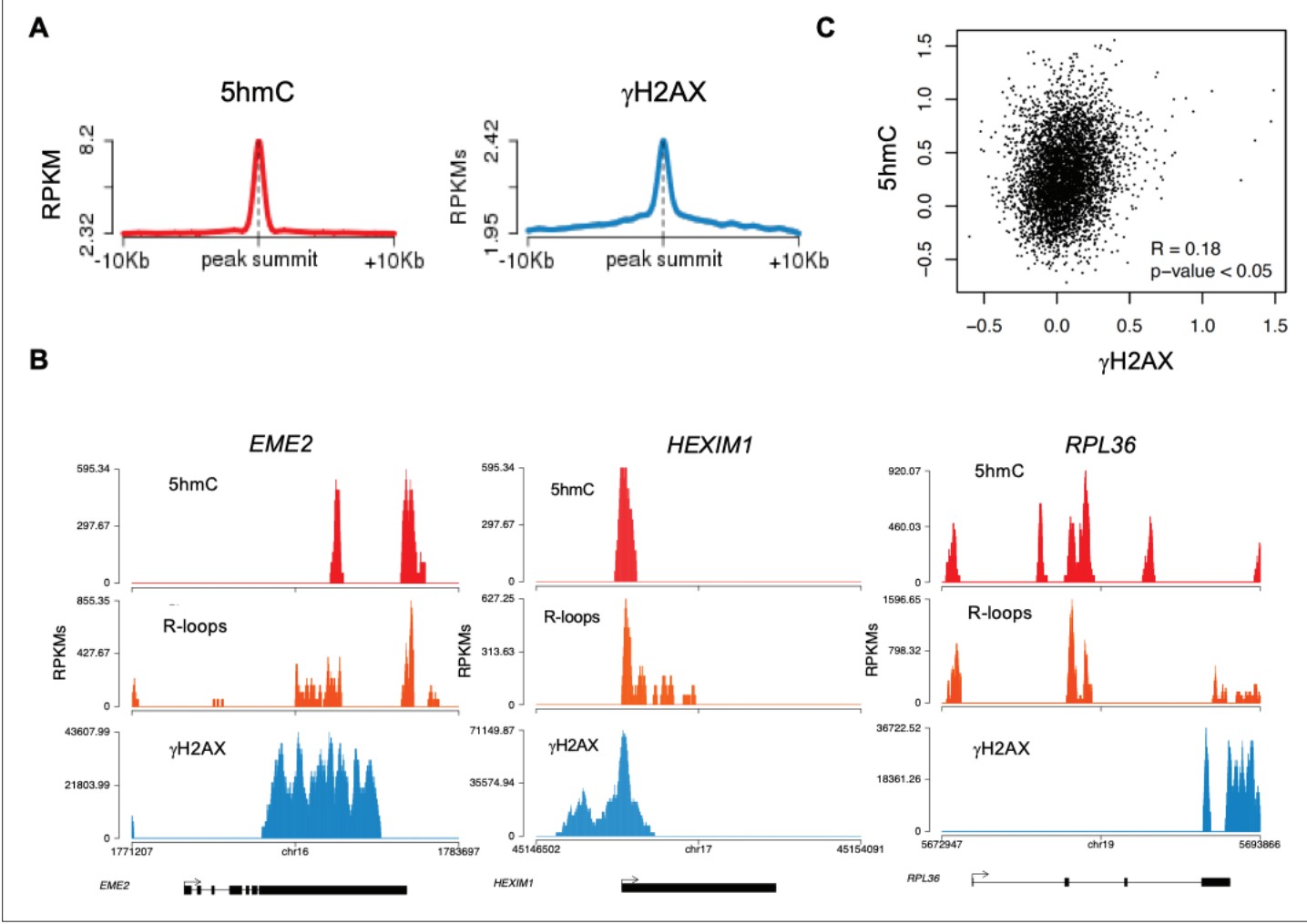

**Figure 5.** 5-hydroxymethylcytosine (5hmC)-rich loci are genomic hotspots for DNA damage. (**A**) Metagene profiles of 5hmC and γH2AX probed over fixed windows of ±10 kbp around the 5hmC peaks in expressed genes of HEK293 cells. (**B**) Individual profiles of 5hmC, R-loops and γH2AX distribution along the *EME2*, *HEXIM1*, and *RPL36* genes. Density signals are represented as reads per kilobase (RPKMs). (**C**) Pearson correlation coefficient between 5hmC and γH2AX at active genes (p<0.05).

expression levels of tumour suppressors or oncogenes (**Bray et al., 2021**), our findings suggest that TET-driven changes in the DNA methylation landscape may as well drive transcription-dependent genome damaging events that could facilitate cancer development and progression. In agreement with this view, a TET1 isoform that lacks regulatory domains, including its DNA-binding domain, but retains its catalytic activity, is enriched in cancer cells (**Good et al., 2017**), suggesting that mistargeted TET activity may drive oncogenic events, such as genomic instability. Conversely, TET activity deposits 5hmC at DNA damage sites induced by aphidicolin or microirradiation in HeLa cells and prevents chromosome segregation defects in response to replication stress (**Kafer et al., 2016**). While the role that TETs play during carcinogenesis is not yet clear, the impact of 5hmC on stem cell differentiation and development has been extensively studied (**Ficz et al., 2011**). By driving the developmental DNA methylome reprogramming, TETs carry out numerous functions related to early developmental processes. Here, we disclose a putative new role for R-loops as mediators of 5hmC-driven gene expression programs in stem cells. Our gene ontology analysis revealed that R-loops formed at 5hmC-rich regions impact the expression of genes involved in establishing diapause. This stage of temporary suspension of embryonic development is triggered by adverse environmental conditions (**Fenelon et al., 2014**). Accordingly, changes in the activity of mTOR, a major nutrient sensor, control ES cell commitment to trigger diapause (**Bulut-Karslioglu et al., 2016**). The mTOR signaling pathway was significantly downregulated upon global R-loop suppression by RNase H. Conversely, MYC targets,

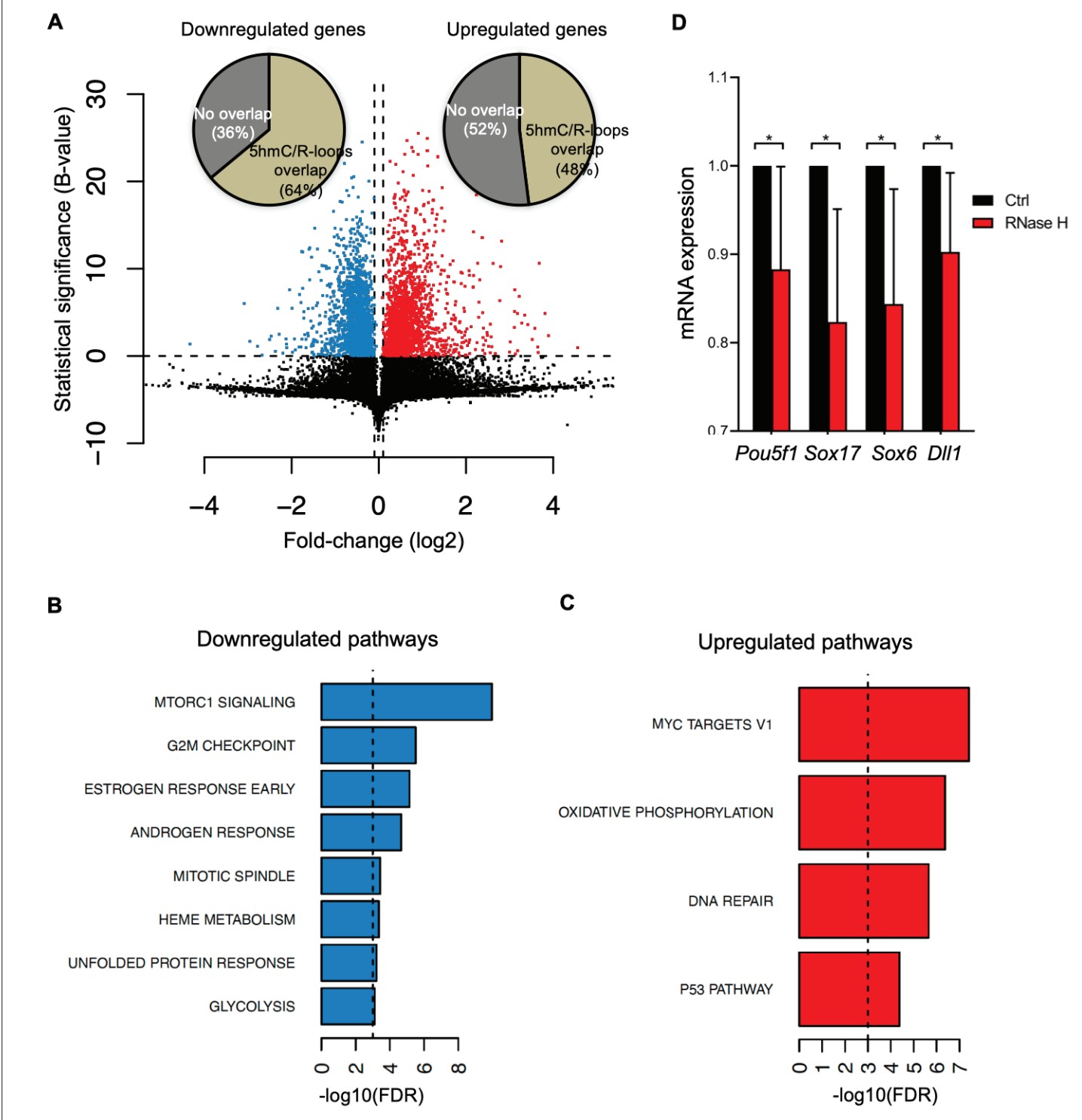

**Figure 6.** Cellular pathways affected by R-loops formed at 5-hydroxymethylcytosine (5hmC) loci. (**A**) Volcano plot displaying the differentially expressed genes in mouse embryonic stem (mES) cells upon RNase H overexpression. Of all downregulated and upregulated genes, 64 and 48% displayed R-loops overlapping with 5hmC, respectively. (**B, C**) Pathway analysis of the genes that have R-loops overlapping with 5hmC and are differentially expressed upon RNase H overexpression. Shown are the significantly downregulated (**B**) and upregulated (**C**) hallmark gene sets from MSigDB. False discovery rate (FDR), p<0.001. (**D**) Transcription levels of pluripotency and germ layer commitment genes in mES cells overexpressing RNase H. *p<0.05, two-tailed Student's *t*-test. Means and SDs are from five independent experiments.

The online version of this article includes the following figure supplement(s) for figure 6:

**Figure supplement 1.** Global R-loop suppression does not impact cell cycle progression of mouse embryonic stem (mES) cells.

which prevent ES cells from entering the state of dormancy that characterizes diapause (*Scognamiglio et al., 2016*), were amongst the genes more significantly upregulated upon RNase H overexpression in mES cells. These cells also displayed reduced expression levels of genes related to pluripotency (*Pou5f1*) and germ layer commitment (Sox17, Sox6, Dll1) pathways. Whether the controlled 5hmC-driven formation of R-loops at specific genes drives stem cells fate and how do TET enzymes capture the environmental cues to target R-loop formation at selected genes are important questions that emerge from our findings. Our study sets the ground for further research aimed at investigating the role of R-loops in ES cells.

## Materials and methods

### Cell lines and culture conditions

E14TG2a (E14) mES cells were provided by Domingos Henrique (Instituto de Medicina Molecular João Lobo Antunes) and were a gift from Austin Smith (University of Exeter, UK) (*Smith and Hooper, 1987*). 129S4/SvJae (J1) mES cells were kindly provided by Joana Marques (Medical School, University of Porto). Cells were grown as monolayers on 0.1% gelatine (410875000, Acros Organics)-coated dishes using Glasgow modified Eagle's medium (GMEM) (21710-025, Gibco), supplemented with 1% (v/v) 200 mM L-glutamine (25030-024, Thermo Scientific), 1% (v/v) 100 mM sodium pyruvate (11360-039, Gibco), 1% (v/v) 100× nonessential amino acids solution (11140-035, Gibco), 0.1% (v/v) 0.1 M 2-mercaptoethanol (M7522, Sigma-Aldrich), 1% (v/v) penicillin-streptomycin solution (15070-063, Gibco), and 10% (v/v) heat-inactivated, ES-qualified FBS (SH30070, Cytiva). Medium was filtered through a 0,22 µm filter. Home-produced leukemia inhibitory factor (LIF) was added to the medium upon plating, at $6 \times 10^{-2}$ ng/µL. U-2 OS osteosarcoma, HEK293T embryonic kidney cells, and NIH-3T3 mouse fibroblasts were purchased from ATCC. Cells were grown as monolayers in Dulbecco's modified Eagle's medium (DMEM) (21969-035, Gibco), supplemented with 1% (v/v) 200 mM L-glutamine (25030-024, Thermo Scientific), 1% (v/v) penicillin-streptomycin solution (15070-063, Gibco), and 10% (v/v) FBS (10270106, Gibco). All cells were maintained at 37°C in a humidified atmosphere with 5% $CO_2$. Cell lines were authenticated using the STR profiling service provided by ATCC and routinely tested negative for mycoplasma contamination using the Mycoplasma Detection Kit (InvivoGen, San Diego, CA).

### *Tet* knockdown

For each *Tet*, a mixture of four siRNAs provided as a single reagent was transfected using Lipofectamine RNAiMAX Transfection Reagent (13778150, Invitrogen) for 48 hr. All siRNAs were purchased as siGENOME SMARTPool from Dharmacon: mouse *Tet1* (M-062861-01), mouse *Tet2* (M-058965-01), and mouse *Tet3* (M-054156-01). A siRNA targeting the firefly luciferase was used as control. For the *Tet1/2/3* triple KD, the three siRNA reagents were combined in the same RNA interference experiment. *Tet3* knockdown was performed in J1 mES cells stably expressing a doxycycline-inducible short hairpin RNA targeting *Tet3* (*Supplementary file 2A*). Cells were treated for 48 hr with 2 µg/mL doxycycline (D9891, Sigma-Aldrich).

### RNA isolation and quantitative RT-PCR

Total RNA was isolated using TRIzol reagent (15596018, Invitrogen). cDNA was prepared through reverse transcriptase activity (MB125, NZYTech). RT-qPCR was performed in the ViiA 7 Real-Time PCR system (Applied Biosystems) using PowerUp SYBR Green Master Mix (A25918, Applied Biosystems). Relative RNA expression was estimated as follows: $2^{(Ct\ reference\ -\ Ct\ sample)}$, where Ct reference and Ct sample are mean threshold cycles of RT-qPCR done in duplicate for *U6* snRNA or *Gapdh* mRNA and for the gene of interest, respectively. Primer sequences are presented in *Supplementary file 2B*.

### Dot blot of genomic R-loops, 5mC, and 5hmC

Cells were lysed in lysis buffer (100 mM NaCl, 10 mM Tris pH 8.0, 25 mM EDTA pH 8.0, 0,5% SDS, 50 µg/mL Proteinase K) overnight at 37°C. Nucleic acids were extracted using standard phenol-chloroform extraction protocol and resuspended in DNase/RNase-free water. Nucleic acids were then fragmented using a restriction enzyme cocktail (20 U each of EcoRI, BamHI, HindIII, BsrgI, and XhoI). Half of the sample was digested with 40 U RNase H (MB085, NZYTech) for 48 hr at 37°C to be used

as a negative control in R-loops blotting. Digested nucleic acids were cleaned with standard phenol-chloroform extraction and resuspended in DNase/RNase-free water. Nucleic acids samples were quantified in a NanoDrop 2000 spectrophotometer (Thermo Scientific), and equal amounts of DNA were deposited into a positively charged nylon membrane (RPN203B, GE Healthcare). Membranes were UV-crosslinked using UV Stratalinker 2400 (Stratagene), blocked in 5% (m/v) milk in PBSt (PBS 1× containing 0.05% [v/v] Tween 20) for 1 hr at room temperature, and immunoblotted with specific antibodies. For the loading control, membranes were stripped in 0.5% SDS for 1 hr at 60°C, followed by blocking and re-probing. Details of the antibodies used are included in *Supplementary file 2C*.

## Proximity ligation assay (PLA)

E14 mES cells were grown on coverslips and fixed/permeabilized with methanol for 10 min on ice, followed by 1 min acetone on ice. Cells were then incubated with primary antibodies for 1 hr at 37°C, followed by a pre-mixed solution of PLA probe anti-mouse minus (DUO92004, Sigma-Aldrich) and PLA probe anti-rabbit plus (DUO92002, Sigma-Aldrich) for 1 hr at 37°C. Localized rolling circle amplification was performed using Detection Reagents Red (DUO92008, Sigma-Aldrich), according to the manufacturer's instructions. Slides were mounted in 1:1000 DAPI in Vectashield. For the RNase H control, fixed cells were treated with 3 U/μL RNase H (MB085, NZYTech) for 1 hr at 37°C prior to incubation with the antibodies. Images were acquired using the Point Scanning Confocal Microscope Zeiss LSM 880, 63×/1.4 oil immersion, with stacking acquisition and generation of maximum intensity projection images. PLA foci per nucleus were quantified using ImageJ. Details of the antibodies used are mentioned in *Supplementary file 2C*.

## g-Blocks PCR

Designed g-blocks were ordered from IDT (*Supplementary file 2D*), and PCR-amplified using Phusion High-Fidelity DNA Polymerase (M0530S, NEB), according to the manufacturer's instructions. M13 primers were used to amplify all fragments (*Supplementary file 2B*) in the presence of dNTP mixes containing native (MB08701, NZYTech), methylated (D1030, Zymo Research), or hydroxymethylated (D1040, Zymo Research) cytosines. Efficient incorporation of modified dCTPs was confirmed through immunoblotting with specific antibodies. Details of the antibodies used are mentioned in *Supplementary file 2C*.

## In vitro transcription

PCR products were subject to in vitro transcription using the HiScribe T7 High Yield RNA Synthesis Kit (E2040S, NEB), which relies on the T7 RNA polymerase to initiate transcription from a T7 promoter sequence (present in our fragments). Reactions were performed for 2 hr at 37°C, using 1 μg of DNA as template, according to the manufacturer's instructions. The resulting RNA was column-purified with NucleoSpin RNA isolation kit (740955.250, Macherey-Nagel) and quantified in a NanoDrop 2000 spectrophotometer (Thermo Scientific).

## Dot blot of R-loops formed in in vitro

Half of each in vitro transcription product was treated with 10 U RNase H (MB085, NZYTech) at 37°C overnight to serve as negative control. Then, all samples were treated with 0.05 U RNase A (10109142001, Roche) at 350 mM salt concentration for 15 min at 37°C and ran on agarose gel. Nucleic acids were transferred overnight to a nylon membrane through capillary transfer. The membrane was then UV-crosslinked twice, blocked in 5% milk in PBSt for 1 hr at room temperature, and incubated with the primary antibody at 4°C overnight. Signal quantification was performed using ImageJ. Details of the antibodies used are included in *Supplementary file 2C*.

## Atomic force microscopy

RNase A-treated in vitro transcription products, treated or not with RNase H, were purified through phenol-chloroform extraction method and resuspended in DNase/RNase-free water. DNA solution was diluted 1:10 in Sigma ultrapure water (with final 10 mM MgCl$_2$) and briefly mixed to ensure even dispersal in solution. A 10 μL droplet was deposited at the center of a freshly cleaved mica disc, ensuring that the pipette tip did not contact the mica substrate. The solution was let to adsorb on mica surface for 1–2 min to ensure adequate coverage. The mica surface was carefully rinsed with Sigma

ultrapure water, so that excess of poorly bound DNA to mica is removed from the mica substrate. Afterward, the mica substrate was dried under a gentle stream of argon gas for approximately 2 min, making sure that any excess water is removed. DNA imaging was performed using a JPK Nanowizard IV atomic force microscope, mounted on a Zeiss Axiovert 200 inverted optical microscope. Measurements were carried out in tapping mode using commercially available ACT cantilevers (AppNano). After selecting a region of interest, the DNA was scanned in air, with scan rates between 0.5 and 0.9 Hz. The setpoint selected was close to 0.3 V. Several images from different areas of the same sample were performed and at least three independent samples for each condition were imaged. All images were of 512 × 512 pixels and analyzed with JPK data processing software.

## CRISPR-assisted 5mC/5hmC genome editing

Lentivirus containing dCas9-TET1 (#84475, Addgene) or dCas9-dTET1 (#84479, Addgene) coding plasmids, as well as one out of three gRNAs (gRNA_1, 2, and 3) coding plasmids designed for the *APOE* last exon, were produced in HEK293T cells co-transfected with the Δ8.9 and VSV-g plasmids using Lipofectamine 3000 Transfection Reagent (L3000015, Invitrogen). After 48 hr, cell culture supernatant was collected and filtered through a 0.45 µm filter. Lentivirus were collected through ultracentrifugation (25,000 rpm, 3 hr, 4°C) using an SW-41Ti rotor in a Beckman XL-90 ultracentrifuge. Virus were resuspended in PBS 1× and stored at –80°C. For infection, a pool of lentivirus containing dCas9-TET1 or dCas9-dTET1, as well as gRNA_1, 2, or 3 coding plasmids, was used to infect seeded U-2 OS cells. After 24 hr, antibiotic selection was performed with 1.5 µg/mL puromycin, and infection proceeded for more 48 hr. 3 days post-infection, cells were harvested and genomic DNA was extracted for subsequent protocols.

## DNA:RNA immunoprecipitation (DRIP)

Cells were collected and lysed in 100 mM NaCl, 10 mM Tris pH 8.0, 25 mM EDTA, 0.5% SDS, 50 µg/mL Proteinase K overnight at 37°C. Nucleic acids were extracted using standard phenol-chloroform extraction protocol and resuspended in DNase/RNase-free water. Nucleic acids were then fragmented using a restriction enzyme cocktail (20 U each of EcoRI, BamHI, HindIII, BsrgI, and XhoI), and 10% of the digested sample was kept aside to use later as input. Half of the remaining volume was digested with 40 U RNase H (MB085, NZYTech) to serve as negative control, for 72 hr at 37°C. Digested nucleic acids were cleaned with standard phenol-chloroform extraction and resuspended in DNase/RNase-free water. DNA:RNA hybrids were immunoprecipitated from total nucleic acids using 5 µg of S9.6 antibody (MABE1095, Merck Millipore) in binding buffer (10 mM $Na_2HPO_4$ pH 7.0, 140 mM NaCl, 0.05% Triton X-100), overnight at 4°C. 50 µL protein G magnetic beads (10004D, Invitrogen) were used to pull down the immune complexes at 4°C for 2–3 hr. Isolated complexes were washed five times (for 1 min on ice) with binding buffer and once with Tris-EDTA (TE) buffer (10 mM Tris pH 8.1, 1 mM EDTA). Elution was performed in two steps, for 15 min at 55°C each, using elution buffer (50 mM Tris pH 8.0, 10 mM EDTA, 0.5% SDS, 60 µg/mL Proteinase K). The relative occupancy of DNA:RNA hybrids was estimated by RT-qPCR as follows: $2^{(Ct\ Input - Ct\ IP)}$, where Ct Input and Ct IP are mean threshold cycles of RT-qPCR done in duplicate for input samples and specific immunoprecipitations, respectively. Data were normalized against the corresponding RNase H-treated samples and plotted as absolute numbers or as fold change over control. Primer sequences are shown in *Supplementary file 2B*.

## 5-(Hydroxy)methylated DNA immunoprecipitation ((h)MeDIP)

Cells were collected and lysed in 100 mM NaCl, 10 mM Tris pH 8.0, 25 mM EDTA, 0.5% SDS, 50 µg/mL Proteinase K overnight at 37°C. Samples were sonicated with four pulses of 15 s at 10 mA intensity using a Soniprep150 sonicator (keeping tubes for at least 1 min on ice between pulses). Fragmented nucleic acids were cleaned with standard phenol-chloroform extraction protocol and resuspended in DNase/RNase-free water. 10% of sample was kept aside to use later as input. The remaining volume was denatured by boiling the samples at 100°C for 10 min, followed by immediate chilling on ice and quick spin. Samples were divided in half, and 5 µg of anti-5mC antibody (61255, Active Motif) or 5 µg of anti-5hmC antibody (39791, Active Motif) were used to immunoprecipitate 5mC and 5hmC, respectively, in binding buffer (10 mM $Na_2HPO_4$ pH 7.0, 140 mM NaCl, 0.05% Triton X-100), overnight at 4°C. 50 µL protein G magnetic beads (10004D, Invitrogen) were used to pull down the immune

complexes at 4°C for 2–3 hr. Isolated complexes were washed five times (for 1 min on ice) with binding buffer and once with TE buffer (10 mM Tris pH 8.1, 1 mM EDTA). Elution was performed in two steps, for 15 min at 55°C each, using elution buffer (50 mM Tris pH 8.0, 10 mM EDTA, 0.5% SDS, 60 µg/mL Proteinase K). The relative occupancy of 5mC and 5hmC was estimated by RT-qPCR as follows: $2^{(Ct_{Input}−Ct_{IP})}$, where Ct Input and Ct IP are mean threshold cycles of RT-qPCR done in duplicate for input samples and specific immunoprecipitations, respectively. Primer sequences are presented in *Supplementary file 2B*.

## Cell cycle analysis

pEGFP-N1 (GFP coding plasmid used as control) was purchased from Addgene, and pEGFP-RNaseH1 (GFP-tagged RNase H1 coding plasmid) was kindly provided by Robert J. Crouch (NIH, USA). Seeded mES cells were transfected with GFP (control) or GFP-tagged RNase H coding plasmids using Lipofectamine 3000 Transfection Reagent (L3000015, Invitrogen). 24 or 48 hr later, cells were trypsinized and pelleted by centrifugation at 500 × *g* for 5 min. Cells were fixed in cold 1% PFA for 20 min at 4°C, followed by permeabilization in 70% ethanol for 1 hr at 4°C. Cells were then treated with 25 µg/mL RNase A (10109142001, Roche) in PBS 1× at 37°C for 20 min, followed by staining with 20 µg/mL propidium iodide (P4864, Sigma-Aldrich) in PBS 1× for 10 min at 4°C. Flow cytometry was performed on a BD Accuri C6 (BD Biosciences), and data were analyzed using FlowJo software.

## Electrophoretic mobility shift assay (EMSA)

DNA:RNA hybrids formed with either C-, 5hmC-, or 5mC-containing DNA were obtained by incubating ssDNA with the complementary ssRNA in annealing buffer (100 mM KAc, 30 mM HEPES pH 7.5). Native and C-modified oligonucleotides were ordered from IDT (*Supplementary file 2E*). Hybrid formation was confirmed in a native polyacrylamide gel. Increasing amounts of S9.6 antibody (MABE1095, Merck Millipore) were added to the DNA:RNA hybrids, and the complexes were ran in a native polyacrylamide gel to assess the S9.6 capacity to bind hybrids containing each of the three C variants. The amount of free probe was quantified using ImageJ.

## Multi-omics data

High-throughput sequencing (HTS) data for mES cells and HEK293 cells were gathered from GEO archive: transcriptome of mES cells (GSE67583); R-loops in mES cells (GSE67581); 5hmC in mES cells (GSE31343); γH2AX in mES cells (GSE69140); active transcription in HEK293 (GRO-seq, GSE51633); R-loops in HEK293 (DRIP-seq, GSE68948); 5hmC modification in HEK293 (hMeDIP-seq, GSE44036); γH2AX (ChIP-seq, GSE75170). Transcriptome profiles of mES cells overexpressing RNase H were obtained from GSE67583. The quality of HTS data was assessed with FastQC (https://www.bioinformatics.babraham.ac.uk/projects/fastqc/).

## 5hmC, R-loop, and γH2AX genome-wide characterization

The HTS datasets produced by immunoprecipitation (DRIP-seq, ChIP-seq, and hMeDIP-seq) were analyzed through the same workflow. First, the reads were aligned to the reference mouse and human genome (mm10 and GRCh38/hg38 assemblies, respectively) with Bowtie (*Langmead et al., 2009*), and filtering for uniquely aligned reads. Enriched regions were identified relative to the input samples using MACS (*Zhang et al., 2008*), with a false discovery rate of 0.05. Finally, enriched regions were assigned to annotated genes, including a 4-kilobase region upstream the TSS and downstream the TTS. Gene annotations were obtained from mouse and human GENCODE annotations (M11 and v23 versions, respectively) and merged into a single transcript model per gene using BEDTools (*Quinlan and Hall, 2010*). For individual and metaprofiles, uniquely mapped reads were extended in the 3′ direction to reach 150 nt with the Pyicos (*Althammer et al., 2011*). Individual profiles were produced using a 20 bp window. For the metaprofiles centered around 5hmC peaks, 5hmc-enriched regions were aligned by the peak summit (maximum of the peak) and the read density for the flanking 10 kbp were averaged in a 200 bp window. For the metagene profiles, the gene body region was scaled to 60 equally sized bins and ±10 kbp gene-flanking regions were averaged in 200 bp windows. All profiles were plotted as normalized reads per kilobase per million mapped reads (RPKMs). A set of in-house scripts for data processing and graphical visualization were written in bash and in the R environmental language (https://www.r-project.org/; *R Core Team, 2018*). SAMtools (*Li et al., 2009*) and BEDTools

were used for alignment manipulation, filtering steps, file format conversion, and comparison of genomic features. Statistical significance of the overlap between 5hmC and R-loops was assessed with enriched regions and permutation analysis. Briefly, random 5hmC and R-loops-enriched regions were generated 1000 times from annotated genes using the shuffle BEDTools function (maintaining the number and length of the originally datasets). The p-value was determined as the frequency of overlapping regions between the random datasets as extreme as the observed.

## Transcriptome analysis

Expression levels (transcripts per million [TPMs]) from RNA-seq and GRO-seq datasets were obtained using Kallisto (*Bray et al., 2016*), where reads were pseudo-aligned to mouse and human GENCODE transcriptomes (M11 and v23, respectively). Transcriptionally active genes for 5hmC and R-loops annotation were defined as those with expression levels higher than the 25th percentile. Differential expression in mES cells overexpressing RNase H was assessed using edgeR (v3.20.9) and limma (v3.34.9) R packages (*Robinson et al., 2010*; *Ritchie et al., 2015*). Briefly, sample comparison was performed using voom transformed values, linear modeling, and moderated *t*-test as implemented in limma R package, selecting significantly differentially expressed genes with B-statistics higher than zero. Significantly enriched pathways of up- and downregulated genes (with overlapping R-loops/5hmC regions) were selected using Fisher's exact test and all expressed genes as background gene list. Evaluated pathways were obtained from the hallmark gene sets of Molecular Signatures Database (MSigDB) (*Liberzon et al., 2015*) and filtered using false discovery rate-corrected p-values<0.05.

For the analysis of transcription readthrough, transcriptome profiles from human embryonic stem cells (WT and *TET1* KO) were obtained from a GEO (GSE169209). RNA-seq data were mapped to the reference human genome (GRCh38) with the STAR v2.7.8a using default parameters (*Dobin et al., 2013*). Transcription readthrough levels were evaluated by counting the number of reads mapping downstream the TTS using ARTDeco (*Roth et al., 2020*) and human genome annotation from GENCODE project (GENCODE release 37). Genes with an enrichment in transcriptional readthrough in *TET1* KO samples relative to the control were identified. Metagene profiles were built using the *computeMatrix* tool from the deepTools v3.5.1 (*Ramírez et al., 2016*) and default packages from Python language. Genes were scaled to equally sized bins of 100 bp so that all annotated TSSs and TTSs were aligned. Regions of 1 kb were added upstream of TSS and downstream of TTS and also averaged in 100 bp bins. All read counts were normalized by the number of mapped reads (RPKM).

## Acknowledgements

We thank our colleagues, Joana Marques, Domingos Henrique, and Robert Crouch, for kind gifts of cell lines, plasmids, and reagents. We thank the technical support and resources provided by the Bioimaging and the Flow Cytometry Facilities of Instituto de Medicina Molecular João Lobo Antunes. This work was funded by PTDC/BIA-MOL/30438/2017 and PTDC/MED-OUT/4301/2020 from Fundação para a Ciência e Tecnologia (FCT), Portugal. Funding was also received from EU Horizon 2020 Research and Innovation Programme (RiboMed 857119). JCS is the recipient of an FCT PhD fellowship PD/BD/128292/2017. Work in CMA's laboratory is supported by "la Caixa" Foundation and FCT, IP (LCF/PR/HP21/52310016; PTDC/BIA-MOL/6624/2020; PTDC/MED-ONC/7864/2020).

## Additional information

### Funding

| Funder | Grant reference number | Author |
|---|---|---|
| Fundação para a Ciência e Tecnologia, Portugal | PTDC/BIA-MOL/30438/2017 | Sérgio Fernandes de Almeida |
| Fundação para a Ciência e Tecnologia, Portugal | PTDC/MED-OUT/4301/2020 | Sérgio Fernandes de Almeida |
| EU Horizon 2020 Research and Innovation Programme | RiboMed 857119 | Sérgio Fernandes de Almeida |

| Funder | Grant reference number | Author |
|---|---|---|
| Fundação para a Ciência e Tecnologia, Portugal | PD/BD/128292/2017 | João C Sabino |
| La Caixa Foundation | LCF/PR/HP21/52310016 | Claus M Azzalin |
| FCT | PTDC/BIA-MOL/6624/2020 | Claus M Azzalin |
| FCT | PTDC/MED-ONC/7864/2020 | Claus M Azzalin |

The funders had no role in study design, data collection and interpretation, or the decision to submit the work for publication.

### Author contributions

João C Sabino, Data curation, Formal analysis, Investigation, Methodology, Writing - review and editing; Madalena R de Almeida, Patrícia L Abreu, Marco M Domingues, Nuno C Santos, Claus M Azzalin, Formal analysis, Methodology, Writing - review and editing; Ana M Ferreira, Paulo Caldas, Formal analysis; Ana Rita Grosso, Formal analysis, Investigation, Methodology, Software, Writing - review and editing; Sérgio Fernandes de Almeida, Conceptualization, Formal analysis, Funding acquisition, Project administration, Supervision, Writing - original draft

### Author ORCIDs

João C Sabino  http://orcid.org/0000-0001-7991-4291
Madalena R de Almeida  http://orcid.org/0000-0001-9539-3289
Patrícia L Abreu  http://orcid.org/0000-0002-6387-8537
Claus M Azzalin  http://orcid.org/0000-0002-9396-1980
Ana Rita Grosso  http://orcid.org/0000-0001-6974-4209
Sérgio Fernandes de Almeida  http://orcid.org/0000-0002-7774-1355

### Decision letter and Author response

Decision letter https://doi.org/10.7554/eLife.69476.sa1
Author response https://doi.org/10.7554/eLife.69476.sa2

## Additional files

### Supplementary files

• Supplementary file 1. Differentially expressed genes upon RNase H overexpression.

• Supplementary file 2. Oligonucleotides and antibodies used in the study. (A) shRNA sequences. (B) Oligonucleotide sequences. (C) Antibodies used in this study. (D) g-Blocks sequences. (E) S9.6 electrophoretic mobility shift assay (EMSA) oligonucleotides.

• Transparent reporting form

• Source data 1. Original, uncropped images of blots and gels.

• Source data 2. Original, uncropped images of blots.

### Data availability

All data generated or analysed during this study are included in the manuscript and supporting files.

The following previously published datasets were used:

| Author(s) | Year | Dataset title | Dataset URL | Database and Identifier |
|---|---|---|---|---|
| Chen PB, Chen HV, Acharya D, Rando OJ | 2015 | R loops regulate promoter-proximal chromatin architecture and cellular differentiation | https://www.ncbi.nlm.nih.gov/geo/query/acc.cgi?acc=GSE67584 | NCBI Gene Expression Omnibus, GSE67583 |

*Continued on next page*

*Continued*

| Author(s) | Year | Dataset title | Dataset URL | Database and Identifier |
|---|---|---|---|---|
| Chen PB, Chen HV, Acharya D, Rando OJ | 2015 | Promoter-proximal R-loops regulate binding of chromatin regulators and pluripotency [DRIP-RNAseq] | https://www.ncbi.nlm.nih.gov/geo/query/acc.cgi?acc=GSE67581 | NCBI Gene Expression Omnibus, GSE67581 |
| Matarese F, Carrillo-de Santa Pau E, Stunnenberg HG | 2011 | 5-hydroxymethylcytosine: the sixth DNA base | https://www.ncbi.nlm.nih.gov/geo/query/acc.cgi?acc=GSE31343 | NCBI Gene Expression Omnibus, GSE31343 |
| Flynn RA, Rubin AJ, Calo E, Bt DO | 2016 | 7SK-BAF axis controls pervasive transcription at enhancers | https://www.ncbi.nlm.nih.gov/geo/query/acc.cgi?acc=GSE69140 | NCBI Gene Expression Omnibus, GSE69140 |
| Liu W, Ma Q, Wong K Li W | 2014 | Brd4 and JMJD6-associated anti-pause enhancers in regulation of transcriptional pause release | https://www.ncbi.nlm.nih.gov/geo/query/acc.cgi?acc=GSE51633 | NCBI Gene Expression Omnibus, GSE51633 |
| Nadel J, Athanasiadou R, Lemetre C, Wijetunga NA, ÓBroin P, Sato H, Zhang Z, Jeddeloh J, Montagna C, Golden A, Seoighe C, Greally J | 2015 | RNA:DNA hybrids in the human genome have distinctive nucleotide characteristics, chromatin composition, and transcriptional relationships | https://www.ncbi.nlm.nih.gov/geo/query/acc.cgi | NCBI Gene Expression Omnibus, GSE68948 |
| Jin C, Lu Y, Jelinek J, Liang S | 2014 | TET1 is a maintenance DNA demethylase that prevents methylation spreading in differentiated cells | https://www.ncbi.nlm.nih.gov/geo/query/acc.cgi?acc=GSE44036 | NCBI Gene Expression Omnibus, GSE44036 |
| Bunch H, Lawney BP, Lin YF, Asaithamby A | 2015 | Transcriptional elongation requires DNA break-induced signalling | https://www.ncbi.nlm.nih.gov/geo/query/acc.cgi?acc=GSE75170 | NCBI Gene Expression Omnibus, GSE75170 |

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
