## [Editor Report]

The study shows a correlation between 5hmC and R loops in mES cells and human HEK293 cells depleted of the TET enzymes Tet1, Tet2 and Tet3 that convert 5mC into 5hmC. The data presented are clearly of significant interest in providing new insight into the potential role of 5hmc DNA in specific transcriptional processes. This implies that 5hmC containing DNA has specific epigenetic features beyond a simple intermediate in interconversion between repressive 5mC and active C DNA.

---

## [Decision Letter]

**Decision letter after peer review:**

Thank you for submitting your article "Epigenetic reprogramming by TET enzymes impacts co-transcriptional R-loops" for consideration by *eLife*. Your article has been reviewed by 3 peer reviewers, one of whom is a member of our Board of Reviewing Editors, and the evaluation has been overseen by Jessica Tyler as the Senior Editor. The reviewers have opted to remain anonymous.

Essential revisions:

1) Figure 1: These in vitro transcription experiments indicate that 5hmC modified DNA has a somewhat greater tendency (at most two-fold) to form R-loops. My serious worry here is that the S9.6 antibody must be biochemically tested to rule out that it has an inherent binding preference for 5hmcDNA:RNA hybrids over either DNA:RNA or 5mcDNA:RNA. Such a preference might explain the apparent selectivity for 5hmC modification in favoring R-loop formation? S9.6 antibody is well known to recognize RNA:RNA as well as RNA:DNA hybrids. Consequently 5hmcDNA:RNA should also be tested. It is hard to evaluate how quantitative is the independent atomic force microscopy method to evaluate R-loop levels. Clearly detecting these blob, spur and loop structures may be somewhat subjective.

2) Figure 2: These data showing either depletion of Tet1 or Tet3 (by RNAi) or the targeted dCAS9 tethering of Tet1 to the R-loop enriched APOE gene, all point to some effect of 5hmcDNA levels on R-loop formation. However again the effects observed are modest (less than 2 fold). Why Tet2 isn't also tested and indeed why a triple knock depletion of Tet_1/2_/3 isn't attempted as the small effects of single Tet mRNA depletions may be due to redundancy between these three mC oxidases.

3) It would be important to see the screen shot of of the whole APOE gene following CRISPR targeting of Tet1 to be sure that the effect is specific for its last exon and that Tet1 tethering is having a localized effect just on this region of the APOE gene.

4) Figure 1 and 2: In order to strengthen the results, it is important to provide statistical significance in the experiment shown. The authors should also demonstrate that 5hmC modification has no impact in the transcription levels in the in vitro model and in APOE gene upon dCAS-TET1 targeting. Considering the effect of 5hmC on the double helix stability, all the observations could be connected to a facilitation of RNA Pol II elongation in the investigated regions. Moreover, a more correct approach of normalization of dot blot signals is recommended. Normalization should be performed on dsDNA signal obtained from the same exact membrane of the signal of interest stripped and probed again.

5) Figure 3: These genomic data are quite compelling. However, the correspondence of 5hmcDNA peaks with TTS R-loops but nor TSS R-loops is surprising and needs further discussion, if not further experimental analysis. For example, does Tet deficiency lead to transcription termination/readthrough defects genome wide?

6) A genome-wide analysis of DNA-RNA hybrids in cells depleted of shTet1 or shTet3 depleted cells is required, so that experimental data could be jointly analyzed with the database to confirm the model and the role of TET in the process defined. Alternatively, a significant number of genes could be validated by DRIP-qPCR to show that TET depletion significantly increases hybrids.

7) A snapshot showing the distribution pattern of one or two long chromosomal regions should be provided to see how coincident are the patterns of R loops and 5hmeC.

8) The metagenomic analysis is made on genes with 5hmC and R loops. However, results should also be shown considering peaks to see a more precise association between better defined regions.

9) It would be important to show that transcription at individual genes identified as R loop accumulating after TET depletion is indeed unaffected. Since TET-depletion causes downregulation and upregulation of genes, it is not possible to make conclusions with just global EU incorporation, in which among other issues, it is unclear the impact on the nucleolus contribute to the signal measured.

10) Figure 4: the quality of the PLA figures needs to be improved. Higher magnification and better resolution in the Z axis are crucial. The signal is not nuclear. Treating cells with a combination of RNAseIII+RNAseT1 might improve specificity of S9.6 antibody. RNaseH control for S9.6 antibody signal should also be provided.

11) Figure 6: The analysis proposed is mainly focused on the contribution of R-loops modulation in mES transcription program, rather than the effect of 5hmC modulation. Since the authors have shown in previous experiments that are able to modulate 5hmC deposition, an interesting result might be obtained by investigating the effect of TET enzymes depletion on mES transcription.

---

## [Author Response]

Essential revisions:1) Figure 1: These in vitro transcription experiments indicate that 5hmC modified DNA has a somewhat greater tendency (at most two-fold) to form R-loops. My serious worry here is that the S9.6 antibody must be biochemically tested to rule out that it has an inherent binding preference for 5hmcDNA:RNA hybrids over either DNA:RNA or 5mcDNA:RNA. Such a preference might explain the apparent selectivity for 5hmC modification in favoring R-loop formation? S9.6 antibody is well known to recognize RNA:RNA as well as RNA:DNA hybrids. Consequently 5hmcDNA:RNA should also be tested. It is hard to evaluate how quantitative is the independent atomic force microscopy method to evaluate R-loop levels. Clearly detecting these blob, spur and loop structures may be somewhat subjective.

The possibility that the S9.6 Ab has an inherent binding preference for 5hmC DNA:RNA hybrids, is indeed very relevant for our study. To address this, we performed an EMSA with increasing Ab:hybrids molar ratio. The quantification of the amount of “free” hybrids in the native gel revealed that the Ab is equally competent to bind DNA:RNA hybrids formed with any of the three C variants. These new data are shown in Figure 1 —figure supplement 1.

R-loop structures in AFM data were defined as in Carrasco-Salas et al.,. The threedimensional architectures resulting from R-loop formation in vitro were quantified in the three experimental conditions (native C, 5hmC and 5mC DNA templates). We now provide a more careful description of these aspects in the revised manuscript.

2) Figure 2: These data showing either depletion of Tet1 or Tet3 (by RNAi) or the targeted dCAS9 tethering of Tet1 to the R-loop enriched APOE gene, all point to some effect of 5hmcDNA levels on R-loop formation. However again the effects observed are modest (less than 2 fold). Why Tet2 isn't also tested and indeed why a triple knock depletion of Tet_1/2_/3 isn't attempted as the small effects of single Tet mRNA depletions may be due to redundancy between these three mC oxidases.

As suggested, we measured the R-loop levels in Tet2-KD and Tet_1/2_/3 triple-KD mouse ES cells (Figure 2A,B). In addition, we measured the levels of R-loops formed in mouse fibroblasts after depletion of each of the three Tet enzymes and after triple-KD experiments (Figure 2 —figure supplement 2C,D). These new data show that the strongest effect on R-loop levels is observed upon the triple KD, supporting the view that there is redundancy between the activity of the three Tet enzymes.

3) It would be important to see the screen shot of of the whole APOE gene following CRISPR targeting of Tet1 to be sure that the effect is specific for its last exon and that Tet1 tethering is having a localized effect just on this region of the APOE gene.

We measured R-loop levels at different regions of the APOE gene, as suggested. The new data are shown in Figure 2F. While the most significant increase in the levels of R-loops was observed at the locus targeted by the dCas9, we also noticed augmented levels throughout the gene. This effect is consistent with the detection of 5hmC along the entire gene and not only at the dCas9-TET1 target locus (Figure 2E). While this may result in part from the lack of resolution of the DRIP assay to isolate distinct regions of the quite small (3,6 kbp) APOE gene, it also raises the hypothesis that R-loops can spread upstream and downstream from their inception site in a zipper-like mechanism that would result in the several hundred base-pair-long structures, which have been detected genome wide in several studies.

4) Figure 1 and 2: In order to strengthen the results, it is important to provide statistical significance in the experiment shown. The authors should also demonstrate that 5hmC modification has no impact in the transcription levels in the in vitro model and in APOE gene upon dCAS-TET1 targeting. Considering the effect of 5hmC on the double helix stability, all the observations could be connected to a facilitation of RNA Pol II elongation in the investigated regions. Moreover, a more correct approach of normalization of dot blot signals is recommended. Normalization should be performed on dsDNA signal obtained from the same exact membrane of the signal of interest stripped and probed again.

To investigate if the 5hmC modification impacts transcription levels in the in vitro model, we measured RNA synthesis from C-, 5hmC- or 5mC-containing DNA templates through RNA column-purification. To test the impact of 5hmC in transcription of the APOE gene upon dCAS-TET1 targeting we quantified RNA synthesis by RT-qPCR. These new data are shown in Figure 1 —figure supplement 2 and Figure 2G. The statistical significance is now provided for all these experiments. These data support our conclusion that increased Rloop levels do not result from increased transcription of 5hmC-marked DNA.

As suggested, the dot blot signals in Figure 2 —figure supplement 1B and Figure 2 —figure supplement 2B are now normalized against the dsDNA signal obtained from the same exact membrane of the signal of interest after stripping and re-probing.

5) Figure 3: These genomic data are quite compelling. However, the correspondence of 5hmcDNA peaks with TTS R-loops but nor TSS R-loops is surprising and needs further discussion, if not further experimental analysis. For example, does Tet deficiency lead to transcription termination/readthrough defects genome wide?

This is a very interesting point and we thank the reviewers for the suggestion. In the revised manuscript we include the results of the analysis of RNA-seq data from human embryonic stem cells. As predicted from the observed correspondence between R-loops and 5hmC at the TTS, we observed increased levels of transcription readthrough in several loci upon TET1 depletion. These novel data are shown in Figure 3G.

6) A genome-wide analysis of DNA-RNA hybrids in cells depleted of shTet1 or shTet3 depleted cells is required, so that experimental data could be jointly analyzed with the database to confirm the model and the role of TET in the process defined. Alternatively, a significant number of genes could be validated by DRIP-qPCR to show that TET depletion significantly increases hybrids.

Following the reviewers’ suggestion, we analyzed R-loops levels in a panel of genes upon Tet_1/2_/3 triple-KD in mouse ES cells. Moreover, we have also performed this DRIP-qPCR assay in mouse fibroblasts. As shown in the new Figure2C and Figure 2 —figure supplement 2E, there is a significant reduction of R-loops levels in Tet-depleted cells. Importantly, this reduction is not caused by diminished transcription levels, as shown in the new Figure 2D and Figure 2 —figure supplement 2F.

7) A snapshot showing the distribution pattern of one or two long chromosomal regions should be provided to see how coincident are the patterns of R loops and 5hmeC.

The distribution pattern of R-loops and 5hmC along two long chromosomal regions are now shown in the new Figure 3 —figure supplement 2.

8) The metagenomic analysis is made on genes with 5hmC and R loops. However, results should also be shown considering peaks to see a more precise association between better defined regions.

The metagenomic analysis was performed using enriched regions from 5hmC and R-loops (i.e. peaks) and assessing their overlap. Then, the overlapping regions were mapped to genes to summarize their occupation in functionally relevant regions. The Methods section was rewritten to clarify this point.

9) It would be important to show that transcription at individual genes identified as R loop accumulating after TET depletion is indeed unaffected. Since TET-depletion causes downregulation and upregulation of genes, it is not possible to make conclusions with just global EU incorporation, in which among other issues, it is unclear the impact on the nucleolus contribute to the signal measured.

We performed RT-qPCR to measure transcription of the genes where we assessed the Rloop levels upon Tet depletion. These data (Figure 2D and Figure 2 —figure supplement 2F) show that Tet activity impacts R-loop levels independently of changes in the transcription rate. Following the reviewers concerns about the global EU incorporation experiments, and since we now measure transcription using RT-qPCR, we have removed EU incorporation data from the revised manuscript.

10) Figure 4: the quality of the PLA figures needs to be improved. Higher magnification and better resolution in the Z axis are crucial. The signal is not nuclear. Treating cells with a combination of RNAseIII+RNAseT1 might improve specificity of S9.6 antibody. RNaseH control for S9.6 antibody signal should also be provided.

We now show PLA images with higher quality. We also include the RNase H control and quantified the PLA foci per nucleus in all experimental conditions. These data are shown in Figure 4 and reveal the specificity of the PLA signal observed.

11) Figure 6: The analysis proposed is mainly focused on the contribution of R-loops modulation in mES transcription program, rather than the effect of 5hmC modulation. Since the authors have shown in previous experiments that are able to modulate 5hmC deposition, an interesting result might be obtained by investigating the effect of TET enzymes depletion on mES transcription.

While we appreciate the relevance of this comment, we hope that the referees agree that investigating the effect of TET enzymes on ES cells transcription goes beyond the immediate scope of our study and constitutes a quite challenging research project on its own. Yet, in this revised manuscript we present additional data further strengthening the hypothesis that R-loops (eventually a 5hmC/R-loops axis) impinge on the regulation of gene expression programs related to pluripotency and germ layer commitment pathways in mES cells (Figure 6D).